# Lower Respiratory Tract Microbiome Signatures of Health and Lung Cancer Across Different Smoking Statuses

**DOI:** 10.3390/cancers17162643

**Published:** 2025-08-13

**Authors:** Vladimir G. Druzhinin, Elizaveta D. Baranova, Pavel S. Demenkov, Liudmila V. Matskova, Alexey V. Larionov, Arseniy E. Yuzhalin

**Affiliations:** 1Department of Genetics and Fundamental Medicine, Kemerovo State University, 650000 Kemerovo, Russia; druzhinin_vladim@mail.ru (V.G.D.); laveivana@mail.ru (E.D.B.); alekseylarionov09@gmail.com (A.V.L.); 2Federal Research Center Institute of Cytology and Genetics, Siberian Branch, Russian Academy of Sciences, 119991 Novosibirsk, Russia; demps@bionet.nsc.ru; 3Department of Microbiology, Tumor and Cell Biology, Karolinska Institutet, 171 77 Stockholm, Sweden; liudmila.matskova@ki.se; 4Department of Molecular and Cellular Oncology, The University of Texas MD Anderson Cancer Center, Houston, TX 77030, USA; 5Research Center for Translational Medicine, Sirius University of Science and Technology, 354340 Sochi, Russia

**Keywords:** tobacco smoking, healthy individuals, lung cancer, sputum, bacterial microbiome, taxonomic composition, microbiome signatures, 16S rRNA, NGS

## Abstract

The lung microbiome represents a dynamic ecological system continuously shaped by both environmental exposures (e.g., tobacco smoke) and host factors (e.g., malignant transformation). It remains incompletely understood how these systemic factors influence the composition and abundance of prokaryotic communities in the respiratory system. In this first large-scale Russian cohort study, we performed a large high-throughput analysis of lower respiratory tract microbiomes in healthy subjects and lung cancer patients of different smoking statuses (current, former, and never smokers). Our findings reveal: (1) distinct smoking-associated microbial signatures persistent across disease states and (2) tumor-specific prokaryotic patterns suggesting cancer-mediated ecological disruption. This study provides new insights into the complex interaction between smoking, oncogenesis, and respiratory microbiome dynamics.

## 1. Introduction

Tobacco smoking is a major global public health problem, contributing to 6–8.7 million deaths annually [1]. Smoking is a primary risk factor for chronic respiratory diseases and exacerbates acute respiratory conditions, including bacterial and viral infections [2,3]. Chronic exposure to tobacco smoke leads to structural changes in the airways [4], reduces tissue oxygenation [5], and impairs alveolar macrophage function [6] and ciliated cell activity [7]. Furthermore, cigarettes contain a wide range of pathogenic bacteria, including *Acinetobacter*, *Bacillus*, *Burkholderia*, *Clostridium*, *Klebsiella*, *Pseudomonas aeruginosa*, and *Serratia* [8,9].

Smoking profoundly influences the bacterial microbiota composition of the lung [10,11]. While numerous studies have documented smoking-induced microbial alterations in the upper respiratory tract [12,13,14,15,16,17,18], less is known about its impact on the lower respiratory tract (LRT). This knowledge gap stems largely from technical challenges in sampling lung tissue, bronchoalveolar lavage, or sputum. Existing LRT studies yield conflicting results. Several studies have reported no significant differences in microbial diversity and composition between smokers and nonsmokers. For instance, Erb-Downward et al. [19] found no major variations in bacterial communities through 16S qPCR analysis of bronchoalveolar lavage samples from seven smokers and three nonsmokers. Similarly, Haldar et al. [20] observed no influence of smoking history on the sputum microbiome in a large cohort of 342 subjects, including individuals with chronic obstructive pulmonary disease (COPD) and healthy controls. Morris et al. [21] also reported minimal differences in the respiratory microbiome of 45 healthy nonsmokers and 19 smokers after sequencing bacterial 16S rRNA genes from oral washes and bronchoscopic alveolar lavages.

In contrast, other studies have identified distinct prokaryotic communities associated with smoking. Einarsson et al. [22] compared the lower airway microbiota among 18 COPD patients, 8 smokers without airway disease, and 11 healthy individuals, revealing significant differences in microbial composition between COPD patients and both smokers and nonsmokers. Similarly, Lim et al. [23] demonstrated that tobacco smoking influences the sputum microbiome in a study of 54 smokers, 170 nonsmokers, and 33 former smokers. Pfeiffer et al. [24] further supported this finding, showing that individual smoking histories alter bacterial community composition in both the upper and lower respiratory tracts of 36 smokers and 15 nonsmokers. Lastly, a large population-based metagenomic analysis by Lin et al. [25] of 1651 healthy subjects confirmed smoking as a key factor shaping airway microbial profiles.

Smoking is also a leading risk factor for lung cancer [26]. Efforts to identify specific microbiome signatures in smoking and nonsmoking lung cancer patients [27,28,29] are limited by small sample sizes and the absence of comparative analyses across healthy individuals and lung cancer patients with varied smoking statuses, including former smokers. Precise LRT microbiome signatures of lung cancer remain to be identified. Understanding why and how smoking and smoking-associated respiratory diseases shape the microbiome is essential for uncovering targetable microbiome–host interactions.

In this study, we conducted genetic profiling of the LRT microbiota in a large cohort of healthy individuals and lung cancer patients, stratified by smoking status (never smokers, former smokers, and current smokers). Our goal was to delineate the individual and combined effects of tobacco exposure and lung cancer on LRT microbiome composition and diversity.

## 2. Methods

### 2.1. Study Groups

Sputum samples were collected from 190 patients with histologically confirmed lung cancer (147 men, 43 women; mean age 61.6 ± 0.59 years), who were admitted to the Kemerovo Regional Oncology Dispensary. A control group included 107 volunteers (88 men, 19 women; mean age 54.7 ± 1.01 years) with no history of airway diseases. Demographic matching between cases (lung cancer patients) and controls (healthy volunteers) was not implemented due to practical recruitment constraints.

Participants completed a survey detailing demographics, residence (urban or rural), occupation, chronic conditions, medication use, and smoking status, which was categorized as current smokers, never smokers (nonsmokers), or former smokers abstinent for at least one year. Smoking intensity was quantified using pack-years. Inclusion criteria were age ≥ 40 years, adequate sputum sample volume, and signed informed consent. Exclusion criteria included acute or chronic conditions that could limit participation, respiratory infections or vaccinations within 3 months prior to sample collection, antibiotic use within four weeks prior to sampling, inability to provide sputum, or refusal to consent. All participants provided informed consent after being briefed on the study’s objectives and risks. The study was approved by the Biomedical Ethics Committee of Kemerovo State University (protocol No. 17/2021, 5 April 2021) and adhered to the principles of the Declaration of Helsinki.

### 2.2. Sputum Collection

Sputum samples (2–3 mL) were collected via productive coughing. To minimize contamination from oral and postnasal microbes, participants cleared their nasal passages and rinsed their mouths before sample collection. No sputum induction methods were used. Samples were immediately transferred to sterile containers, frozen at −20 °C, transported to the laboratory, and stored at −80 °C until DNA extraction. Several randomly selected sputum samples were Giemsa-stained to microscopically confirm the presence of columnar airway epithelial cells.

### 2.3. DNA Extraction, Library Preparation, Sequencing, and Microbiome Analysis

Prokaryotic DNA was extracted using the FastDNA Spin Kit for Soil (MPbio, Solon, OH, USA) following the manufacturer’s protocol. DNA concentration and quality were assessed using a Qubit^®^ fluorometer (Life Technologies, Carlsbad, CA, USA). DNA samples with a concentration > 15 ng/µL, volume > 20 µL, and A260/A280 ratio of 1.8–2.0 were used for library preparation.

The 16S rRNA gene was amplified following the Illumina protocol for “Preparing 16S Ribosomal RNA Gene Amplicons for the Illumina MiSeq System.” Amplicons (~500 bp) targeting the hypervariable V3–V4 regions of the 16S rRNA gene were generated using broad-spectrum primers. A second PCR step with indexed primers produced fragments (~630 bp). The MiSeq V3 reagent kit (Illumina, San Diego, CA, USA) was used for sequencing. Negative controls without biological material were included during sample preparation and processed identically to study samples. ZymoBIOMICS™ microbial community standard (Zymo Research, Irvine, CA, USA) was used as a positive control. The following amplification primers for 16S rRNA genes were used: forward: 5′-TCGTCGGCAGCGTCAGATGTGTATAAGAGACA-GCCTACGGGNGGCWGCAG-3′; reverse: 5′-GTCTCGTGGGCTCGGAGATGTGTATAAGAGAC-AGGACTACHVGGGTATCTAATCC-3′.

DNA polymerase BioMaster Hi-Fi LR 2 × ReadyMix (Biolabmiks, Novosibirsk, Russia) was used for amplification. PCR conditions were: 94 °C for 3 min 30 s; 25 cycles at 94 °C (30 s), 55 °C (30 s), and 68 °C (40 s); followed by 68 °C for 5 min and cooldown at 4 °C. Libraries (~550 bp) were purified using Agencourt AMPure XP beads (Beckman Coulter, Brea, CA, USA). Indexed adapters from Illumina Nextera XT v2 kits (Illumina, San Diego, CA, USA) were added in a second PCR. Amplified libraries were purified and quantified using a Quantus Fluorometer (Promega, Madison, WI, USA). Final libraries were prepared at 4 nM, denatured, diluted to 8 pM, followed by addition of 10% PhiX DNA. The quality of the final libraries was assessed using the Bioanalyzer 2100 (Agilent, Santa Clara, CA, USA) with the Agilent DNA 1000 Kit. Samples that failed quality control during either DNA extraction or library preparation were excluded from the experiment. Finally, samples were sequenced using the MiSeq system using the MiSeq Reagent Kit v2 (300 cycles). Raw read counts were converted to relative abundance (proportions) for community composition analysis. For alpha/beta diversity metrics, we performed rarefaction to the minimum sequencing depth (1000 reads/sample) to ensure equal sampling effort across comparisons. Average sequencing depth was 11,431.

### 2.4. Microbiome Taxonomy and Statistical Analysis

Sequencing data were analyzed using the QIIME 2 software [30]. Amplicons were quality-checked, and sequences were clustered into operational taxonomic units (OTUs) at a 99% similarity threshold using Greengenes (v13.8) and SILVA (v138) reference databases. Singleton OTUs were removed.

Alpha diversity was assessed via OTU richness, Shannon index (i.e., measure of α-diversity, taking into account the number of taxa (species) and their evenness of distribution in the community), and Pielou evenness index. Beta diversity was analyzed using UniFrac metrics [31], and samples were normalized by 1070 sequences (minimum of sequences obtained per sample). PERMANOVA (Adonis) was used to assess differences between groups, taking into account interactions between all taxa.

Linear discriminant analysis effect size (LEfSe) identified biologically relevant bacterial taxa that differ significantly between groups [32], estimating both the significance (*p*-value) and the effect size (LDA score), which allows identification of the most important taxa. Mann–Whitney U tests evaluated relative bacterial abundances, with significance set at *p* < 0.05. False discovery rate (FDR) corrections were used to account for multiple comparisons. Unique sequencing reads were compared between groups using two-way ANOVA with uncorrected Fisher’s least significant difference post hoc test. Spearman’s correlation evaluated the relationship between bacterial abundance and smoking intensity. Statistical analyses were performed using STATISTICA 10 (Statsoft, Tulsa, OK, USA).

## 3. Results

### 3.1. Next-Generation Sequencing of Human Sputum Samples

To investigate the composition of the airway microbiome, we noninvasively collected sputum samples from 107 healthy individuals without a history of airway conditions and 197 patients with lung cancer. The demographic and clinicopathological characteristics of the study participants are summarized in Table 1. As expected, lung cancer patients exhibited a higher prevalence of comorbidities, such as cardiovascular and chronic obstructive pulmonary diseases, compared to healthy individuals (*p* < 0.05, Table 1). Each study group was further stratified into nonsmokers, smokers, and former smokers. As expected, lung cancer patients had significantly higher smoking intensity compared to healthy smokers (35.3 vs. 27.2 pack-years, *p* = 0.0018, Table 1). Histologically, adenocarcinoma mostly affected nonsmokers whereas smokers were more frequently diagnosed with squamous cell carcinoma (*p* = 0.02, Table 1), a pattern consistent with previous reports [33]. Following prokaryotic DNA extraction from all sputum samples (N = 297), the 16S ribosomal RNA (rRNA) gene was amplified, and next-generation sequencing was performed (Figure 1A).

Sequencing of the V3–V4 region of the 16S rRNA gene identified nine bacterial phyla with a relative abundance greater than 0.1% in sputum samples. The predominant phyla were Firmicutes and Bacteroidetes, accounting for approximately 70% of the total microbiota (Appendix A). A total of 43 bacterial genera were identified with a frequency of at least 0.1% (Figure 1B,C, Appendix A), among which the most abundant were *Streptococcus*, *Prevotella*, *Veillonella*, and *Anaerosinus*. We then compared the number of unique sequences reads between groups as an indirect measure of microbial richness and relative abundance. Healthy nonsmokers exhibited the highest cumulative number of unique reads among all study groups (Figure 1D), with a borderline significance (*p* = 0.05) compared to healthy smokers. However, among lung cancer patients, the number of unique reads did not significantly differ between smokers and nonsmokers (Figure 1D).

### 3.2. Impact of Smoking on LRT Microbiota of Healthy Subjects

First, we questioned whether smoking alters the LRT microbiome of healthy individuals. Bacterial composition showed no significant differences in alpha diversity in healthy subjects, irrespective of smoking status (Figure 2A). Similarly, beta diversity (measured as distance between nonsmokers, smokers, and former smokers) was not different in healthy individuals (Figure 2B). However, comparison of bacterial abundance revealed major differences between groups (Appendix A). *Streptococcus* (phylum Firmicutes) was the only bacterium significantly enriched in the sputum of healthy smokers (22.36% vs. 16.55%, *p* < 0.01) compared to healthy nonsmokers (Figure 2C,D), indicating that tobacco smoking creates a permissive environment for *Streptococcus*-associated conditions such as bronchitis and pneumonia. Conversely, the phyla Proteobacteria and Fusobacteria, as well as the genera *Neisseria*, *Fusobacterium,* and *Alloprevotella*, were significantly reduced in the sputum of healthy smokers compared to nonsmokers (Figure 2C,D, Appendix A). Interestingly, the LRT microbiome of healthy former smokers more closely resembled that of healthy nonsmokers than current smokers. For example, the abundance of *Neisseria* in the sputum of former smokers was 5.39%, similar to that of healthy nonsmokers (4.49%), and significantly higher than in current smokers (1.04%, *p* < 0.05, Figure 2D). Compared with healthy nonsmokers and healthy former smokers who both displayed a high abundance of *Neisseria*, healthy smokers demonstrated very low abundance of this bacterium (Figure 2E). These data indicate that certain native lung microbiome populations are sensitive to tobacco smoke exposure but can recover over time following smoking cessation.

### 3.3. Impact of Smoking on LRT Microbiota of Lung Cancer Patients

Lung cancer in combination with smoking induces major respiratory stress. In contrast to healthy individuals, sputum microbiota of smoking lung cancer patients showed a reduced alpha diversity compared to both nonsmoking patients (H = 6.701, *p* = 0.009) and former smokers (H = 3.493, *p* = 0.0061), suggesting a weakened microbial ecosystem of subjects experiencing extreme respiratory distress (Figure 2F). Yet, lung cancer patients showed no significant differences in beta diversity depending on smoking status as demonstrated by minimal distances between study cohorts (Figure 2G). The prokaryotic content in the LRT of lung cancer patients was overall similar between smokers and nonsmokers, with few exceptions (Appendix A). *Neisseria* was significantly less frequent in smokers compared to nonsmokers (2.86% vs. 3.67%; *p* = 0.01, Figure 2H, Appendix A), similarly to findings observed in healthy individuals (Appendix A). This reduction was observed in both adenocarcinoma patients (3.89% vs. 4.84%, *p* = 0.03) and squamous cell carcinoma patients (2.7% vs. 4.24%, *p* = 0.04) (Appendix A). Furthermore, differences in *Neisseria* were mostly affecting smokers at later stages (III–IV) of lung cancer compared to nonsmoking patients (2.86% vs. 4.24%; *p* = 0.02, Figure 2I).

In addition, smoking lung cancer patients exhibited a significantly higher abundance of *Selenomonas* (1.82% vs. 1.18%; *p* = 0.03) and, conversely, a decrease in *Fusobacterium* (1.49% vs. 1.95%; *p* = 0.04) compared to nonsmoking patients (Figure 2H). However, these differences did not remain statistically significant after FDR adjustment for multiple comparisons.

Interestingly, the LRT microbiome of lung cancer patients who had quit smoking (n = 12) generally showed no substantial differences from that of smokers or nonsmokers. Notable exceptions included an increase in *Macellibacteroides* (3.58% vs. 1.66%; *p* = 0.02) and *Lachnoanaerobaculum* (0.59% vs. 0.27%; *p* = 0.01) genera in the sputum of former smokers compared to smoking lung cancer patients (Figure 2H, Appendix A). These findings suggest that lung cancer exerts a greater influence on the composition of LRT microbiota than smoking alone and that smoking cessation during lung cancer may not be sufficient to reshape microbiome diversity to baseline levels.

### 3.4. Comparison of LRT Microbiome Between Lung Cancer Patients and Healthy Individuals, Irrespective of Smoking Status

We then directly compared the LRT microbiota of lung cancer patients and healthy controls, without considering smoking status. Alpha diversity of the prokaryotic communities did not differ between lung cancer patients and healthy controls (Figure 3A). In contrast, the Pielou evenness index (Figure 3B) was significantly higher in healthy individuals (*p* = 0.0013) compared to cancer patients, suggesting that the normal LRT microbiome represents a balanced ecosystem with no single species dominating excessively. In addition, beta diversity analysis revealed significant compositional differences in the respiratory microbiome between lung cancer patients and healthy controls (pseudo-F = 3.07; *p* = 0.001, Figure 3C). This finding suggests that lung cancer is associated with distinct microbial community structures, potentially reflecting disease-induced ecological disruptions or host–microbe interactions specific to the tumor microenvironment.

The abundance of bacteria belonging to dominant phyla (Firmicutes, Bacteroidetes, and Proteobacteria) was not significantly different between lung cancer patients and healthy subjects (Figure 3D). However, six less common phyla (Actinobacteria, Fusobacteria, TM7, Spirochaetes, SR1, and Tenericutes) were significantly reduced in the LRT microbiome of cancer patients compared to healthy individuals (Figure 3D).

To better understand microbiome changes associated with lung cancer independent of smoking status, we employed the linear discriminant analysis effect size (LEfSe) method. Integrating statistical significance testing with effect size estimation, this approach revealed a significant enrichment of specific prokaryotic taxa in the LRT of lung cancer patients compared to healthy controls (Figure 3E), including members of the phylum Proteobacteria; classes Bacilli, Gammaproteobacteria, and Flavobacteriia; orders Lactobacillales, Pasteurellales, Bacillales, and others (Figure 3F).

Conversely, the sputum of healthy individuals was dominated by representatives of less common bacterial taxa, including *Clostridia*, *Coriobacteria*, *Alphaproteobacteria*, and others (Figure 3F). Collectively, these findings highlight a substantial transformation in the lung microbiome by lung cancer, independent of tobacco exposure.

### 3.5. Comparison of LRT Microbiomes Between Lung Cancer Patients and Healthy Individuals, Dependent on Smoking Status

We then investigated how tobacco exposure shapes the LRT microbiota in healthy individuals and lung cancer patients. We found no difference in alpha diversity between smoking lung cancer patients and healthy smokers (Figure 4A), indicating comparable overall microbial diversity within individual respiratory niches. However, beta diversity analysis revealed pronounced compositional differences between these groups (pseudo-F = 2.099, *p* = 0.001, Figure 4B,C), suggesting that lung cancer is associated with structural reorganization of the microbiome rather than changes in overall diversity. The LEfSe analysis revealed major differences in the LRT microbiome composition of smoking lung cancer patients compared to smoking healthy individuals. Specifically, the LRT of smokers with lung cancer was enriched in bacteria belonging to the classes Flavobacteriia, Gammaproteobacteria, and Bacilli; orders Flavobacteriales, Pasteurellales, Bacillales, Lactobacillales, and others. In contrast, healthy smokers exhibited overrepresentation of prokaryotes from the classes Alphaproteobacteria and Coriobacteria; orders Clostridiales, Coriobacteriales, Fusobacteriales, and others (Figure 4D).

There was no difference in alpha diversity between nonsmoking lung cancer patients and healthy nonsmokers (Figure 5A), whereas beta diversity displayed significant compositional difference between these groups (pseudo-F = 1.664, *p* = 0.0012, Figure 5B), indicating that, while overall microbial diversity was similar, the specific community structure differed substantially. The sputum of nonsmoking lung cancer patients exhibited an increased abundance of bacteria from the class Flavobacteriia; orders Flavobacteriales, Pasteurellales, and Bacillales; families Pasteurellaceae, Bacillaceae, Aerococcaceae, Porphyromonadaceae, and Weeksellaceae; and genera *Bacillus*, *Granulicatella*, *Macelibacteroides*, *Bergeyella*, and *Peptostreptococcus* (Figure 5C). Conversely, the microbiome of healthy nonsmokers was enriched with representatives from the phylum Saccharibacteria (TM7); class Alphaproteobacteria; families Microbacteriaceae and Coriobacteriaceae; and genera *Clostridium*, *Spirochaeta*, *Shutieworthia*, and *Moryella* (Figure 5C).

We then analyzed the microbiomes of lung cancer patients and healthy subjects who quit smoking. There was no difference in alpha or beta diversity between former smokers in both groups (Figure 5D,E), whereas the LEfSe analysis revealed a significant increase in the abundance of bacteria belonging to the phylum Bacteroidetes; class Flavobacteriia; orders Flavobacteriales and Lactobacillales; families Bacillaceae, Flavobacteriaceae, Clostridiales, Leptotrichiaceae, etc. in the LRT of former smokers with lung cancer (Figure 5F). In contrast, the sputum microbiome of healthy subjects who quit smoking was enriched with the family Veillonellaceae and genus *Anaerosinus* (Figure 5F).

### 3.6. Abundance of Selenomonas in the LRT Correlates with Smoking Exposure in Lung Cancer

Finally, we examined the relationship between cumulative smoking exposure (pack-years) and LRT microbiome abundance using Spearman’s correlation analysis (Appendix A). Among smokers with lung cancer, only one significant positive correlation was observed, linking increased smoking intensity with high abundance of the genus *Selenomonas* (r = 0.2563, *p* = 0.0074, Figure 6A). In contrast, the sputum microbiome of healthy smokers exhibited exclusively negative correlations. Specifically, higher smoking intensity was significantly associated with reduced abundance of *Campylobacter* (r = −0.3829, *p* = 0.0079), *Lachnoanaerobaculum* (r = −0.3427; *p* = 0.0184), and *Peptostreptococcus* (r = −0.4122, *p* = 0.004) (Figure 6B). The contrasting patterns between smoking lung cancer patients and healthy smokers suggest that certain bacterial taxa in the LRT microbiome respond differently to smoking in the presence of cancer. This could reflect the altered immune landscape, tumor microenvironment, or disease-driven changes in microbial selection pressures in lung cancer patients.

## 4. Discussion

Our study is the first comprehensive characterization of respiratory microbiota in a Russian population, comparing healthy individuals and lung cancer patients across smoking status groups. The Russian context may influence microbiome profiles through several unique factors: the prevalence of particular tobacco products (including higher consumption of distinct local blends), environmental exposures (such as extreme seasonal climate variations and urban air pollution patterns in major cities), and population-specific lifestyle factors including dietary habits and antibiotic usage patterns.

Our findings suggest a strong influence of tobacco smoking on LRT microbiota composition and diversity. In healthy individuals, smoking led to increased abundance of *Streptococcus* (phylum Firmicutes) and reduced Proteobacteria and Fusobacteria and genera *Fusobacterium*, *Alloprevotella*, *Capnocytophaga*, *Zhouea*, and *Neisseria*. Notably, former smokers’ microbiota closely resembled that of nonsmokers, rather than that of current smokers. These results align with a previous study [34], which reported similar microbiome shifts in smokers’ oropharyngeal samples but noted trends, such as increased *Megasphaera*, *Veillonella*, *Actinomyces*, and *Atopobium*, not observed here. Furthermore, elevated *Streptococcus* and reduced *Neisseria* levels were reported in a study of the oral microbiome of American adults [17]. Similarly, a multicenter cohort study compared the upper and lower respiratory tract microbiomes of healthy smokers and nonsmokers [21], reporting increased *Streptococcus* and Fusobacterium and decreased *Porphyromonas*, *Neisseria*, and *Gemella* in smokers’ oropharyngeal samples. Clearly, these changes reflect oxygen-deprivation effects in smokers’ airways favoring facultatively anaerobic species such as *Streptococcus* over aerobic ones, such as *Neisseria*. Smoking’s interference with nitrate metabolism could also contribute to this difference, as reduced nitrate levels in saliva and airways are linked to smoking [35,36].

In our study, sputum from healthy smokers showed a marked reduction in *Neisseria* compared to nonsmokers and former smokers. Increased smoking duration and intensity correlated with declines in other genera, such as *Campylobacter*, *Lachnoanaerobaculum*, and *Peptostreptococcus,* in healthy smokers. Existing data on sputum microbiomes in healthy smokers and nonsmokers are sparse, with only a few publications available. For instance, a study on Korean twins revealed smoking’s influence on sputum microbiota composition, with increased *Veillonella* and *Megasphaera* and decreased *Haemophilus* abundance as smoking intensity increased [23].

In patients with lung cancer, microbiota diversity between smokers, former smokers, and nonsmokers was minimal. This suggests that tumor-associated factors may dominate over smoking history in shaping microbial communities, likely because the systemic inflammatory and immunomodulatory effects of malignancy exert a more profound influence on the respiratory microbiome than smoking-related changes alone. This phenomenon aligns with emerging evidence that cancer creates a pervasive microenvironment that fundamentally alters host–microbe interactions across anatomical sites. Nonetheless, *Neisseria* abundance was consistently lower in smokers, particularly in advanced disease stages (III–IV). Only a few bacteria, such as *Neisseria*, maintain a significant association with smoking even in lung cancer patients. A lack of dramatic variation in the LRT microbiome composition between smoking and nonsmoking lung cancer patients has also been reported in other studies [37].

Analysis of sputum microbiota between lung cancer patients and healthy donors independent of smoking status revealed distinct microbiome signatures. Whereas *Flavobacteriia*, *Bacillales*, and *Pasteurellales* were enriched in lung cancer patients’ LRT, *Alphaproteobacteria*, *Coriobacteriaceae*, and *Microbacteriaceae* were reduced. Smoking-specific microbiota differences included increased *Gammaproteobacteria* and *Lactobacillales* in smokers with lung cancer and reduced *Fusobacteriales*, *Clostridiales*, *Veillonellaceae*, and *Leptotrichiaceae*. Interestingly, the elevated levels of *Streptococcus* in the respiratory tract of lung cancer patients have been reported in previous studies [28,38,39] and were proposed as a potential biomarker for lung cancer [40]; however, in our study *Streptococcus* was not identified as a lung cancer-specific bacterium after taking into account smoking status.

This study has several limitations. Whereas the sputum analyzed in our study serves as a surrogate for the LRT microbiota, it may still be contaminated with bacteria from the upper respiratory tract during sample collection [41]. Also, this study did not test for associations of LRT microbiome with lung cancer survival or molecular subtypes of lung cancer [42].

## 5. Conclusions

In summary, we established microbiome signatures of the LRT (Figure 7) in a large cohort of healthy individuals and lung cancer patients of different smoking statuses. Smoking significantly alters respiratory microbiota in healthy individuals, with notable effects on *Streptococcus*, *Alloprevotella*, *Capnocytophaga*, *Zhouea*, and *Neisseria*. In lung cancer, these effects diminish, likely due to influence of the tumor microenvironment. Smoking status should be carefully considered in microbiome research to avoid misinterpretation.

## Figures and Tables

**Figure 1 cancers-17-02643-f001:**
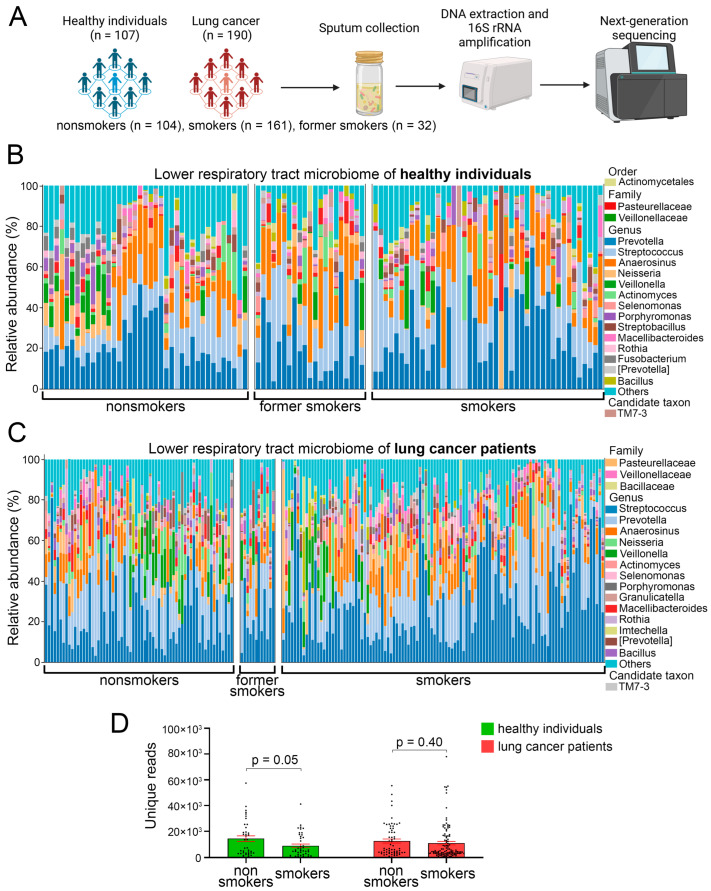
Study design and overview of LRT microbiome of healthy individuals and lung cancer patients across different smoking statuses. (**A**) Schematic of the experimental design. Created with biorender.com. (**B**,**C**) Most prevalent bacterial families and genera found in sputum samples of healthy individuals (n = 107) (**B**) and lung cancer patients (n = 190) (**C**) with different smoking status (**D**). The total number of unique sequencing reads identified in sputum samples of healthy smokers (n = 48), healthy nonsmokers (n = 39), smoking lung cancer patients (n = 113), and nonsmoking lung cancer patients (n = 65). Two-way ANOVA with uncorrected Fisher’s least significant difference. Each dot indicates a study subject. Line indicates mean, error bars indicate standard error of the mean.

**Figure 2 cancers-17-02643-f002:**
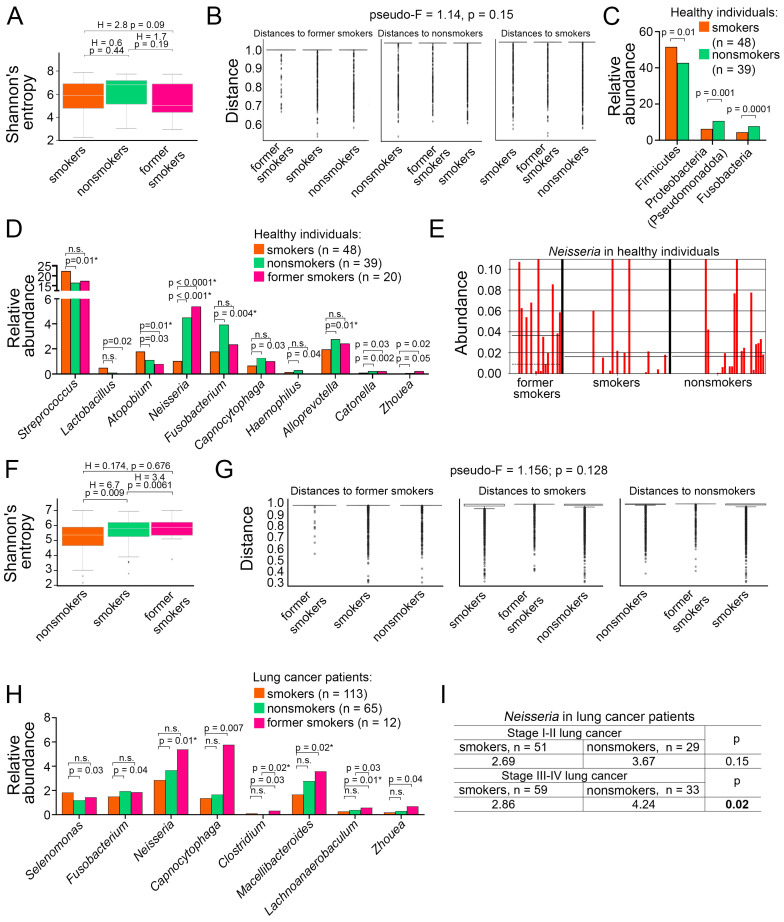
Impact of smoking on LRT microbiota of healthy individuals and lung cancer patients. (**A**) The Shannon diversity indices of LRT microbiome of healthy individuals across different smoking statuses. Kruskal–Wallis test. Line indicates median, boxes indicate 1st and 3rd quartiles, whiskers indicate range. (**B**) The beta diversity of LRT microbiome, measured as distance, between healthy individuals across different smoking statuses. PERMANOVA (Adonis) test. Line indicates median, boxes indicate 1st and 3rd quartiles, dots indicate outliers. (**C**) Significantly different phyla in the sputum of healthy individuals across different smoking statuses. Mann–Whitney U test with FDR correction for multiple comparisons. (**D**) Significantly different genera in the LRT microbiome of healthy individuals across different smoking statuses. Mann–Whitney U test with FDR correction for multiple comparisons. * *p*-value is less than FDR-adjusted *p*-value. (**E**) Relative abundance of *Neisseria* in the sputum of healthy former smokers (n = 20), smokers (n = 48), and nonsmokers (n = 39). Solid line indicates mean, dashed line indicates median. (**F**) The Shannon diversity indices of LRT microbiome of lung cancer patients across different smoking statuses. Kruskal–Wallis test. Line indicates median, boxes indicate 1st and 3rd quartiles, whiskers indicate range, dots indicate outliers. (**G**) The beta diversity of LRT microbiomes, measured as distance, between lung cancer patients across different smoking statuses. PERMANOVA (Adonis) test. Line indicates median, boxes indicate 1st and 3rd quartiles, dots indicate outliers. (**H**) Significantly different genera in the LRT microbiome of lung cancer patients across different smoking statuses. Mann–Whitney U test with FDR correction for multiple comparisons. * *p*-value is less than FDR-adjusted *p*-value. (**I**) Relative abundance of *Neisseria* in the sputum of smokers and nonsmokers in early-stage (I, II) and late-stage (III, IV) lung cancer. Mann–Whitney U test. LRT, lower respiratory tract; FDR, false discovery rate; n.s., non-significant.

**Figure 3 cancers-17-02643-f003:**
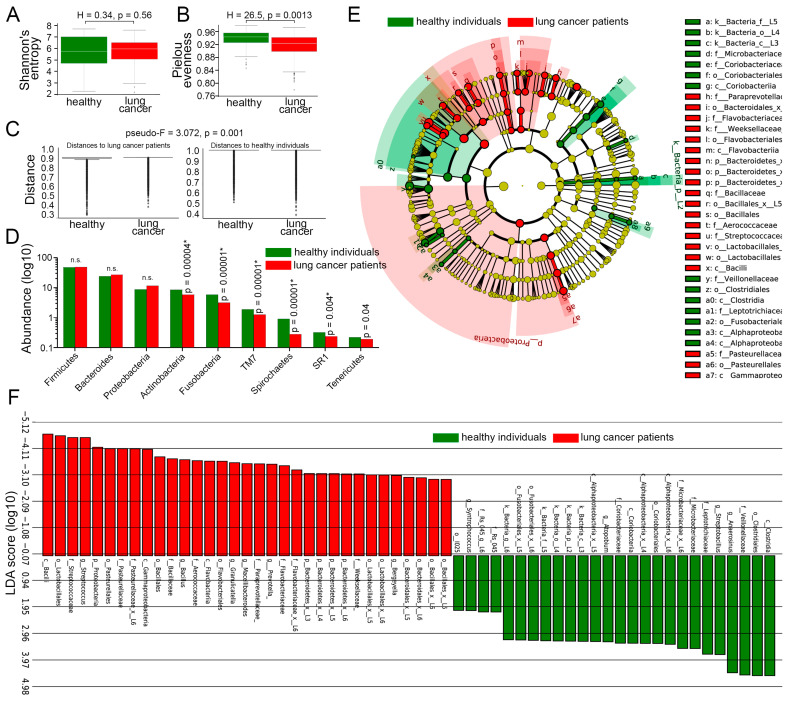
Differences in LRT microbiota of healthy subjects and lung cancer patients, disregarding smoking status. (**A**) The Shannon diversity indices of LRT microbiomes of healthy individuals (n = 107) and lung cancer patients (n = 190) regardless of smoking status. Mann–Whitney U test. Line indicates median, boxes indicate 1st and 3rd quartiles, whiskers indicate range, dots indicate outliers. (**B**) Pielou index of LRT microbiomes of healthy individuals (n = 107) and lung cancer patients (n = 190) regardless of smoking status. Mann–Whitney U test. Line indicates median, boxes indicate 1st and 3rd quartiles, whiskers indicate range, dots indicate outliers. (**C**) The beta diversity of LRT microbiomes, measured as distance, between healthy individuals (n = 107) and lung cancer patients (n = 190) regardless of smoking status. PERMANOVA (Adonis) test. Line indicates median, boxes indicate 1st and 3rd quartiles, dots indicate outliers. (**D**) Abundance of major phyla in LRT microbiomes of healthy individuals (n = 107) and lung cancer patients (n = 190) regardless of smoking status. Mann–Whitney U test with FDR correction for multiple comparisons. * *p*-value is less than FDR-adjusted *p*-value. (**E**) A cladogram showing evolutionary relationships among different bacterial populations in LRT microbiomes of healthy individuals (n = 107) and lung cancer patients (n = 190) independent of smoking status. (**F**) LEfSe analysis demonstrating LRT microbiome differences between healthy individuals (n = 107) and lung cancer patients (n = 190) regardless of smoking status. LEfSe, linear discriminant analysis effect size; LRT, lower respiratory tract; FDR, false discovery rate; n.s., non-significant.

**Figure 4 cancers-17-02643-f004:**
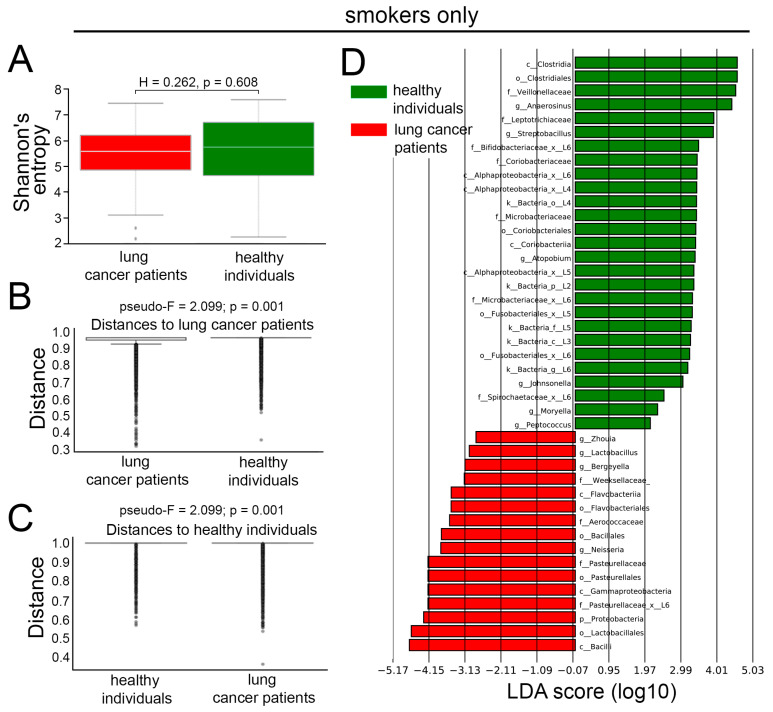
Differences in LRT microbiota of healthy smokers and smokers with lung cancer. (**A**) The Shannon diversity indices of LRT microbiomes of healthy smokers (n = 48) and smokers with lung cancer (n = 113). Mann–Whitney U test. Line indicates median, boxes indicate 1st and 3rd quartiles, whiskers indicate range, dots indicate outliers. (**B**,**C**) The beta diversity of LRT microbiomes, measured as distance, between healthy smokers (n = 48) and smokers with lung cancer (n = 113). PERMANOVA (Adonis) test. Line indicates median, boxes indicate 1st and 3rd quartiles, dots indicate outliers. (**D**) LEfSe analysis demonstrating LRT microbiome differences between healthy smokers (n = 48) and smokers with lung cancer (n = 113). LEfSe, linear discriminant analysis effect size; LRT, lower respiratory tract.

**Figure 5 cancers-17-02643-f005:**
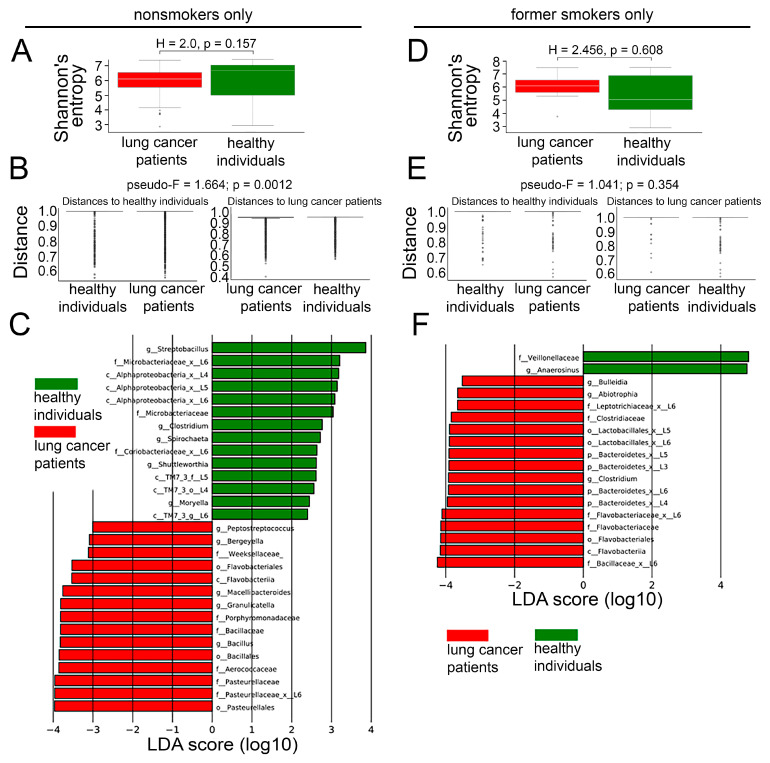
LRT microbiota differences between healthy nonsmokers and nonsmokers with lung cancer, and between healthy former smokers and former smokers with lung cancer. (**A**) The Shannon diversity indices of LRT microbiomes of nonsmoking healthy individuals (n = 39) and nonsmoking lung cancer patients (n = 65). Mann–Whitney U test. Line indicates median, boxes indicate 1st and 3rd quartiles, whiskers indicate range, dots indicate outliers. (**B**) The beta diversity of LRT microbiomes, measured as distance, between nonsmoking healthy individuals (n = 39) and nonsmoking lung cancer patients (n = 65). PERMANOVA (Adonis) test. Line indicates median, boxes indicate 1st and 3rd quartiles, dots indicate outliers. (**C**) LEfSe analysis of LRT microbiome differences between healthy nonsmoking healthy individuals (n = 39) and nonsmoking lung cancer patients (n = 65). (**D**) The Shannon diversity indices of LRT microbiomes of healthy former smokers (n = 20) and lung cancer patients who quit smoking (n = 12). Mann–Whitney U test. Line indicates median, boxes indicate 1st and 3rd quartiles, whiskers indicate range, dots indicate outliers. (**E**) The beta diversity of LRT microbiomes, measured as distance, between healthy former smokers (n = 20) and lung cancer patients who quit smoking (n = 12). PERMANOVA (Adonis) test. Line indicates median, boxes indicate 1st and 3rd quartiles, dots indicate outliers. (**F**) LEfSe analysis of LRT microbiome differences between healthy former smokers (n = 20) and lung cancer patients who quit smoking (n = 12). LEfSe, linear discriminant analysis effect size; LRT, lower respiratory tract.

**Figure 6 cancers-17-02643-f006:**
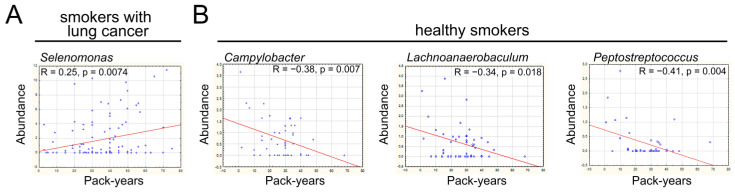
Correlation between smoking intensity and bacterial abundance. (**A**) The abundance of *Selenomonas* positively correlates with smoking intensity in lung cancer patients (n = 113). Spearman’s correlation analysis. (**B**) The abundance of *Campylobacter*, *Lachnoanaerobaculum*, and *Peptostreptococcus* negatively correlates with smoking intensity in healthy smokers (n = 48). Spearman’s correlation analysis.

**Figure 7 cancers-17-02643-f007:**
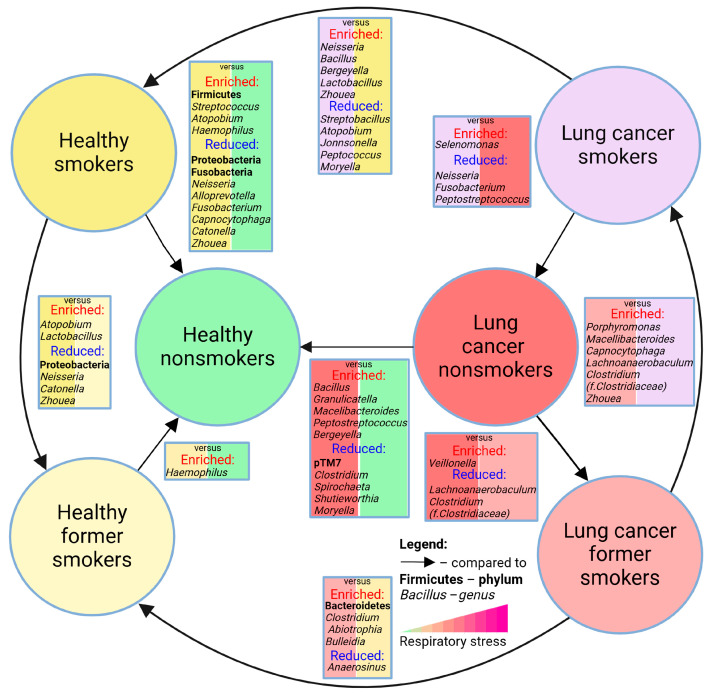
Cartoon summarizing sputum bacterial signatures (phyla and genera) of healthy individuals and lung cancer patients across different smoking statuses. Created with biorender.com.

**Table 1 cancers-17-02643-t001:** Clinicopathological characteristics of the study groups.

Baseline Characteristics	Lung Cancer Patients, n = 190			Healthy Individuals, n = 107			*p*-Value *
	Smokers, n = 113	Nonsmokers, n = 65	Former Smokers, n = 12	Smokers, n = 48	Nonsmokers, n = 39	Former Smokers, n = 20	
Age, years	61.3	61.9	62.7	53.3	55.8	56.2	0.416
Gender (n/%):							
Male	101/89.6	36/55.4	12/100	45/93.7	26/66.7	17/82.2	
Female	12/10.6	29/44.6	-	3/6.3	13/33.3	3/15.0	2.53 × 10^−8^
Chronic conditions (n/%):							
Cardiovascular disease	49/43.4	39/60.0	10/83.3	7/14.6	14/35.9	5/25.0	1.52 × 10^−6^
Bronchitis	22/19.5	10/15.4	3/25.0	-/0	-/0	-/0	0.000266
COPD	27/23.9	5/7.7	2/16.7	-/0	-/0	-/0	3.29 × 10^−6^
Gastrointestinal disease	12/10.6	9/13.9	-/0	9/18.8	4/10.3	2/10.0	0.512
Diabetes	4/3.5	8/12.3	-/0	1/2.1	2/5.1	1/5.0	0.127
Asthma	-/0	3/4.6	2/16.7	-/0	1/2.6	-/0	0.00192
Obesity	6/5.3	7/10.8	-/0	-/0	-/0	-/0	0.0344
Pack-years	35.3 ± 15.7			27.2 ± 12.9			0.0018
Histological subtype of lung cancer (n/%):							
Squamous cell carcinoma	49/43.4	14/21.5	3/25.0				
Adenocarcinoma	38/33.6	33/50.8	4/33.3				
Others	26/23.0	18/27.7	5/41.7				0.0289
TNM (n/%):							
I, II	43/38.1	30/46.2	3/25.0				
III, IV	70/61.9	35/53.8	9/75.0				0.3120

Abbreviations: COPD, Chronic obstructive pulmonary disease; TNM, Tumor, node, metastasis. * χ^2^ test was used for all comparisons, excluding age (Kruskal–Wallis test) and pack-years (Mann–Whitney U test).

## Data Availability

Microbiome sequencing data that support the findings of this study have been deposited in the BioProject under accession code PRJNA1219364. All other data supporting the findings of this study are available from the corresponding author on reasonable request.

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
