# Peer review of "Lower Respiratory Tract Microbiome Signatures of Health and Lung Cancer Across Different Smoking Statuses"

_cancers, 2025, doi:10.3390/cancers17162643_

Round 1

Reviewer 1 Report

Comments and Suggestions for Authors

Dear Editor and Authors,

It was of interest to read and review this manuscript titled "Lower Respiratory Tract Microbiome Signatures of Health and Lung Cancer Across Different Smoking Statuses".

This is an interesting and uniques I might say concept, which although not sure how it could be applicable in clinical practice it does pose interesting questions and querries!

In this work the authors collected sputum samples and characterized the lower respira-
tory tract microbiome in a patient cohort of 297 individuals from Russia which included both healthy subjects and lung cancer patients of different smoking statuses (current smokers, former smokers and nonsmokers). Their findings were analyzed and compared.

Methodologically this is a sound study with clear parameters and inclusion/exclusion criteria. In terms of molecular analysis correct procedures and tests/examinations were performed utilizing appropriate methods. The sample size of patients is adequate to produce statistically meaningful results.

The results are quite interesting and open avenues for further investigation and discussion. They are well presented and discussed. The tables and images included are quite informative, clear and descriptive.

The discussion is extensive, poignant and quite good. 

In conclusion, this is quite a good manuscript and I support its publication. Thank you.

Comments on the Quality of English Language

Needs some minor language editing and proofreading.

Author Response

Reviewer 1.

Comment:  Dear Editor and Authors,

It was of interest to read and review this manuscript titled "Lower Respiratory Tract Microbiome Signatures of Health and Lung Cancer Across Different Smoking Statuses". This is an interesting and uniques I might say concept, which although not sure how it could be applicable in clinical practice it does pose interesting questions and querries! In this work the authors collected sputum samples and characterized the lower respiratory tract microbiome in a patient cohort of 297 individuals from Russia which included both healthy subjects and lung cancer patients of different smoking statuses (current smokers, former smokers and nonsmokers). Their findings were analyzed and compared.

Methodologically this is a sound study with clear parameters and inclusion/exclusion criteria. In terms of molecular analysis correct procedures and tests/examinations were performed utilizing appropriate methods. The sample size of patients is adequate to produce statistically meaningful results.

The results are quite interesting and open avenues for further investigation and discussion. They are well presented and discussed. The tables and images included are quite informative, clear and descriptive.

The discussion is extensive, poignant and quite good.  In conclusion, this is quite a good manuscript and I support its publication. Thank you.

Response:
We express sincere thanks to the reviewer for carefully reading and appreciating our work.

Reviewer 2 Report

Comments and Suggestions for Authors

Important and interesting study.

Introduction

“Tobacco smoking is a major global public health problem, contributing to 6–8.7 mil- 43
lion deaths annually(Perez-Warnisher et al., 2018)”

[Reference now 7 years out of date. Please update]

“Furthermore, cigarettes contain a wide range of pathogenic bacteria, including Acinetobacter, Bacillus, Burkholderia, Clostridium, Klebsiella, Pseudomonas aeruginosa, and Serratia(Sapkota et al., (2010).”

[Please specify country of origin of tobacco and cigarettes and number of cigarettes sampled. Do you have a more up to date reference?]

“Existing LRT studies yield conflicting results: some report no significant differences in microbial diversity and composition between smokers and nonsmokers(Erb-Downward et al., 2011; Haldar et al., 2020; Morris et al., 2013; Pfeiffer et al., 2022), whereas others identify distinct prokaryotic communities associated with smoking(Campos et al., 2023; Einarsson et al., 2016; Lim et al., 2016; Lin et al., 2023).”

[Please describe sample sizes, research design and risk of bias of studies]

Methods

“A control group included 107 volunteers (88 men, 19 women; mean age 54.7 ± 1.01 years) with no history of airway diseases.”

[Please describe how you selected this sample and effect of small sample of females (19) on conclusions you can draw. Do you have a history of respiratory infections and samples during these infections so you have a colonisation origin history?]

[Do you have vaccination histories. Important especially for pneumococcus].

“Prokaryotic DNA was extracted using the FastDNA Spin Kit for Soil (MPbio, USA)”

[Did you use the soil kit as you are searching for the origin of the microbiomes in the soil the tobacco was grown in?]

Results

Please reconsider all your graphics and whether readers will be able to read the small print. Can the readers understand your results clearly and promptly from the graphics? I do not think so.

Need explanation for Fig 2 Panels B, E and G.

Explanation of Fig 3 C; Fig 4 B and C; Fig 5 B and E; Figure 6

Explain Shannon’s entropy in detail

Author Response

Reviewer 2.

Comment: Important and interesting study.

Response:
We express sincere thanks to the reviewer for carefully reading and appreciating our work.

Comment:  Introduction

“Tobacco smoking is a major global public health problem, contributing to 6–8.7 mil- 43 lion deaths annually(Perez-Warnisher et al., 2018)” [Reference now 7 years out of date. Please update]

Response:

We have replaced this obsolete reference with a more recent one:

Fu M, Mei A, Min X, Yang H, Wu W, Zhong J, Li C, Chen J. Advancements in Cardiovascular Disease Research Affected by Smoking. Rev Cardiovasc Med. 2024;25(8):298.

Comment: “Furthermore, cigarettes contain a wide range of pathogenic bacteria, including Acinetobacter, Bacillus, Burkholderia, Clostridium, Klebsiella, Pseudomonas aeruginosa, and Serratia(Sapkota et al., (2010).”

[Please specify country of origin of tobacco and cigarettes and number of cigarettes sampled. Do you have a more up to date reference?]

Response:

We appreciate the reviewer’s insightful comment regarding the origin of the tobacco and the number of cigarettes sampled in the cited study (Sapkota et al., 2010). Unfortunately, in our study, we were unable to specify the country of origin or manufacturer of the cigarettes used by participants, as this information was not systematically collected. Participants sourced cigarettes based on individual preferences, affordability, and availability, introducing variability that was difficult to account for in our analysis.

Regarding the request for a more recent reference, we acknowledge that additional studies have since expanded on the microbial contamination of tobacco products. For instance, Tyx et al. (2016) provides further evidence on this topic. However, we retained Sapkota et al. (2010) as it remains a key foundational study in this area. Thus, in addition to Sapkota et al. (2010) we added the reference by Tyx et al. (2016): Tyx RE, Stanfill SB, Keong LM, Rivera AJ, Satten GA, Watson CH. Characterization of Bacterial Communities in Selected Smokeless Tobacco Products Using 16S rDNA Analysis. PLoS One. 2016;11(1):e0146939.

Comment: “Existing LRT studies yield conflicting results: some report no significant differences in microbial diversity and composition between smokers and nonsmokers(Erb-Downward et al., 2011; Haldar et al., 2020; Morris et al., 2013; Pfeiffer et al., 2022), whereas others identify distinct prokaryotic communities associated with smoking(Campos et al., 2023; Einarsson et al., 2016; Lim et al., 2016; Lin et al., 2023).”

[Please describe sample sizes, research design and risk of bias of studies]

Response: In the abovementioned studies, sample sizes were as follows:

  1. Erb-Downward et al., 2011 – 7 smokers and 3 non-smokers
  2. Haldar et al., 2020 – 124 healthy subjects (28 smokers and 96 non-smokers)
  3. Morris et al., 2013 – 64 healthy subjects (19 smokers and 45 non-smokers)
  4. Pfeiffer et al., 2022 – 52 healthy subjects (36 smokers and 15 non-smokers)
  5. Einarsson et al., 2016 – 18 adults with COPD, 8 smokers with no airways disease and 11 healthy individuals.
  6. Lim et al., 2016 – 257 healthy subjects (54 smokers and 170 non-smokers, 33 former-smokers)
  7. Lin et al., 2023 – 1651 healthy subjects

Notably, none of the above studies assessed the risk of bias.

We have now expanded our introduction section as follows:

“Several studies have reported no significant differences in microbial diversity and composition between smokers and nonsmokers. For instance, Erb-Downward et al. (2011) found no major variations in bacterial communities through 16S qPCR analysis of bronchoalveolar lavage samples from seven smokers and three nonsmokers. Similarly, Haldar et al. (2020) observed no influence of smoking history on the sputum microbiome in a large cohort of 342 subjects, including individuals with chronic obstructive pulmonary disease (COPD) and healthy controls. Morris et al. (2013) also reported minimal differences in the respiratory microbiome of 45 healthy nonsmokers and 19 smokers after sequencing bacterial 16S rRNA genes from oral washes and bronchoscopic alveolar lavages.

In contrast, other studies have identified distinct prokaryotic communities associated with smoking. Einarsson et al. (2016) compared the lower airway microbiota among 18 COPD patients, 8 smokers without airways disease, and 11 healthy individuals, revealing significant differences in microbial composition between COPD patients and both smokers and nonsmokers. Similarly, Lim et al. (2016) demonstrated that tobacco smoking influences the sputum microbiome in a study of 54 smokers, 170 nonsmokers, and 33 former smokers. Pfeiffer et al. (2022) further supported this finding, showing that individual smoking histories alter bacterial community composition in both the upper and lower respiratory tracts of 36 smokers and 15 nonsmokers. Lastly, a large population-based metagenomic analysis by Lin et al. (2023) of 1,651 healthy subjects confirmed smoking as a key factor shaping airway microbial profiles”.

Comment: Methods “A control group included 107 volunteers (88 men, 19 women; mean age 54.7 ± 1.01 years) with no history of airway diseases.”

[Please describe how you selected this sample and effect of small sample of females (19) on conclusions you can draw. Do you have a history of respiratory infections and samples during these infections so you have a colonisation origin history?]

[Do you have vaccination histories. Important especially for pneumococcus].

Response: Thank you for your constructive question. The study cohort was randomly selected based on participants’ informed consent and adherence to the predefined inclusion criteria. While the proportion of women differed between the control group (17.8%) and the patient group (23%), this disparity did not influence the study’s conclusions, as the analysis accounted for potential confounding variables, including sex.

Additionally, to minimize the impact of recent respiratory illnesses or immune responses, we excluded individuals with respiratory infections or vaccinations within three months prior to sample collection (please see page 3, lines 114-115). This criterion helped ensure that the observed outcomes were not biased by acute infections or recent immunological interventions.

Comment: “Prokaryotic DNA was extracted using the FastDNA Spin Kit for Soil (MPbio, USA)”

[Did you use the soil kit as you are searching for the origin of the microbiomes in the soil the tobacco was grown in?]

Response:  We appreciate this question regarding our DNA extraction protocol. While the FastDNA Spin Kit for Soil is indeed marketed for soil samples, it can also be used for other types of samples. As per manufacturer’s instructions: “The FastDNA Spin Kit for Soil is designed to isolate bacterial, fungal, plant, and animal genomic DNA from soil and other environmental samples.” We selected it based on its proven efficacy for challenging biological samples, as demonstrated in prior microbiome studies (Zhu L, Ke H, Wang Q, Xu K, Chen X. Multi-omics profiling reveals distinct pathogenic mechanisms in Hunner and non-Hunner interstitial cystitis subtypes. Sci Rep. 2025 Jul 22;15(1):26536. doi:10.1038/s41598-025-12010-w). The kit's rigorous mechanical lysis and inhibitor removal steps are particularly effective for breaking tough cell walls of environmental and pathogenic bacteria, removing PCR inhibitors common in respiratory samples (e.g., mucins, inflammatory proteins), and achieving high DNA yields from low-biomass samples.

Therefore, this choice was not related to tobacco growth conditions, but rather to optimal DNA recovery from our clinical specimens. Microbiome studies have successfully used this kit (Zhu L, Ke H, Wang Q, Xu K, Chen X. Multi-omics profiling reveals distinct pathogenic mechanisms in Hunner and non-Hunner interstitial cystitis subtypes. Sci Rep. 2025 Jul 22;15(1):26536. doi:10.1038/s41598-025-12010-w). We acknowledge this could be clarified in the Methods section and are happy to add explanatory text if preferred.

Comment: Results Please reconsider all your graphics and whether readers will be able to read the small print. Can the readers understand your results clearly and promptly from the graphics? I do not think so.

Response:

We appreciate this important feedback regarding the clarity and readability of our figures. We have carefully reconsidered all graphical presentations and implemented the following comprehensive improvements to enhance accessibility of our results:

  1. All axis labels, legends, and annotations have been enlarged across all figures, in addition, we increased font sizes for statistical markers (p-values) and taxonomic labels;
  2. We removed excessive white space through tighter figure margins, and rebalanced panel sizes to prioritize data visualization over empty space;
  3. We divided the original Figure 6 into two separate figures (now Figure 6 and Figure 7). This allows each figure to occupy full-width display with larger graphical elements.

We believe these changes substantially improve the reader experience and we're grateful for the opportunity to enhance our visual presentation. Please find the updated figures in the revised submission.

Comment: Need explanation for Fig 2 Panels B, E and G.

Response: Thank you for your comment. We have now provided a detailed explanation of Fig 2 Panels B, E and G in the manuscript text:

Similarly, beta diversity (measured as distance between nonsmokers, smokers, and former smokers) was not different in healthy individuals (Figure 2B).

Compared with healthy nonsmokers and healthy former smokers who both displayed a high abundance of Neisseria, healthy smokers demonstrated very low abundance of this bacterium (Figure 2E).

Yet, lung cancer patients showed no significant differences in beta diversity depending on smoking status as demonstrated by minimal distances between study cohorts (Figure 2G).

All these changes are highlighted in green color in the manuscript text.

Comment: Explanation of Fig 3 C; Fig 4 B and C; Fig 5 B and E; Figure 6

Response: Thank you for your comment. We have now provided a detailed explanation of Fig 3 C; Fig 4 B and C; Fig 5 B and E; Figure 6 in the manuscript text:

“In addition, beta diversity analysis revealed significant compositional differences in the respiratory microbiome between lung cancer patients and healthy controls (pseudo-F = 3.07; p = 0.001, Figure 3C). This finding suggests that lung cancer is associated with distinct microbial community structures, potentially reflecting disease-induced ecological disruptions or host-microbe interactions specific to the tumor microenvironment.”

“However, beta diversity analysis revealed pronounced compositional differences between these groups (pseudo-F = 2.099, p = 0.001, Figure 4B, C), suggesting that lung cancer is associated with structural reorganization of the microbiome rather than changes in overall diversity.”

“…whereas beta diversity displayed significant compositional difference between these groups (pseudo-F = 1.664, p = 0.0012, Figure 5B), indicating that while overall microbial diversity was similar, the specific community structure differed substantially.”

“Among smokers with lung cancer, only one significant positive correlation was observed, linking increased smoking intensity with high abundance of the genus Seleno-monas (r = 0.2563, p = 0.0074, Figure 6A). In contrast, the sputum microbiome of healthy smokers exhibited exclusively negative correlations. Specifically, higher smoking intensity was significantly associated with reduced abundance of Campylobacter (r = -0.3829, p = 0.0079), Lachnoanaerobaculum (r = -0.3427; p = 0.0184), and Peptostreptococcus (r = -0.4122, p = 0.004) (Figure 6B).”

All these changes are highlighted in green color in the manuscript text.

Comment: Explain Shannon’s entropy in detail

Response: Thank you for your comment. Shannon entropy is a measure of α-diversity used in ecology and bioinformatics to quantify biological diversity. It takes into account the number of taxa (species) and their evenness of distribution in the community. The formula is:

H=−∑i=1S​pi​ln(pi​)

where:

  1. H is Shannon entropy,
  2. pi ​ is the relative frequency of the i-th taxon (e.g., species or phylotype),
  3. S is the total number of unique taxa,
  4. ln is the natural logarithm.

Low entropy (H≈0) indicates that the community consists of a single dominant taxon (low diversity). High entropy (H>3) suggests that many taxa are distributed approximately equally (high diversity).

To address the reviewer’s question, the following information was added to the Methods section:

Alpha diversity was assessed via OTU richness, Shannon index (i.e. measure of α-diversity, taking into account the number of taxa (species) and their evenness of distribution in the community), and Pielou evenness index.

All these changes are highlighted in green color in the manuscript text.

We sincerely thank the reviewer for their constructive comments which greatly improved our manuscript!

Reviewer 3 Report

Comments and Suggestions for Authors

This manuscript presents a well-structured and thorough investigation into the effects of smoking and lung cancer on the lower respiratory tract (LRT) microbiome using 16S rRNA sequencing of unstimulated sputum samples from a large Russian cohort. The study is scientifically sound, addresses an important knowledge gap, and uses appropriate methodologies. The stratification by smoking status across health and disease states is particularly valuable.

However, several areas would benefit from revisions to improve clarity, interpretability, and impact. These include streamlining results, enhancing the discussion of statistical methods, and refining the presentation of figures and tables.

Major Comments:

  1. Clarity and Redundancy in the Results:

    • The results are detailed but at times repetitive—particularly in descriptions of taxa like Neisseria. Condense overlapping findings and emphasize overarching patterns.

    • Consider re-structuring results around key comparisons (e.g., smoking vs. non-smoking, health vs. disease), which may improve narrative flow.

  2. Statistical Methods:

    • Further detail is needed regarding data normalization, control of batch effects, and the rationale for selected thresholds (e.g., minimum read counts).

    • The manuscript would benefit from briefly explaining why PERMANOVA and LEfSe were chosen and their assumptions.

  3. Presentation of Figures:

    • Ensure all figures, especially complex plots like LEfSe cladograms, are legible when printed. Improve label clarity in the cartoon summary (Figure 6C).

    • Refer to each figure in the main text at the first relevant mention to help guide the reader.

  4. Interpretation and Discussion:

    • Expand the discussion on how the Russian cohort context may affect microbiome profiles and the generalizability of findings.

    • Add more depth when interpreting why cancer seems to mask the microbiome effects of smoking.

    • Clarify the biological relevance of correlations (e.g., Selenomonas and smoking intensity) with hypotheses for mechanism if available.

  5. Sampling and Methodological Details:

    • Provide more detail on the demographic matching between control and patient groups.

    • Discuss potential contamination of sputum samples by upper airway microbiota, especially since this is a known limitation.

Minor Comments:

  • Abstract: Briefly mention key statistical methods and highlight the inclusion of former smokers as a strength.

  • Line 20: Change “Here, we characterize…” to “We characterized…” for consistency in tense.

  • Line 259: Rephrase “may not fully restore microbiome diversity” to clarify timeframe/context.

  • Review references for consistency in formatting (some DOI entries are inconsistent).

  • Consider mentioning the sequencing depth and whether rarefaction was performed.

Author Response

Reviewer 3.

This manuscript presents a well-structured and thorough investigation into the effects of smoking and lung cancer on the lower respiratory tract (LRT) microbiome using 16S rRNA sequencing of unstimulated sputum samples from a large Russian cohort. The study is scientifically sound, addresses an important knowledge gap, and uses appropriate methodologies. The stratification by smoking status across health and disease states is particularly valuable.

However, several areas would benefit from revisions to improve clarity, interpretability, and impact. These include streamlining results, enhancing the discussion of statistical methods, and refining the presentation of figures and tables.

Major Comments:

Comment: Clarity and Redundancy in the Results: The results are detailed but at times repetitive—particularly in descriptions of taxa like Neisseria. Condense overlapping findings and emphasize overarching patterns. Consider re-structuring results around key comparisons (e.g., smoking vs. non-smoking, health vs. disease), which may improve narrative flow.

Response: Thank you for your valuable comment. To improve clarity and eliminate redundancy, we restructured the Results section to remove non-essential information. The new optimized text is highlighted in yellow color in the revised manuscript.

Comment: Statistical Methods: Further detail is needed regarding data normalization, control of batch effects, and the rationale for selected thresholds (e.g., minimum read counts). The manuscript would benefit from briefly explaining why PERMANOVA and LEfSe were chosen and their assumptions.

Response: Thank you for this technical question. Raw read counts were converted to relative abundance (proportions) for community composition analysis. For alpha/beta diversity metrics, we performed rarefaction to the minimum sequencing depth (1,000 reads/sample) to ensure equal sampling effort across comparisons.

PERMANOVA test was chosen for the following reasons:

  • Independence of observations: samples within groups are independent of each other (e.g., patients are not related genetically or geographically).
  • Appropriate distance metric: the metric used (e.g., Bravais-Curtis, Jaccard, Ward) adequately reflects the differences between samples.
  • No assumptions of normality: PERMANOVA is a nonparametric test based on permutations, making it robust to violations of normality.
  • PERMANOVA assesses how much the microbiome structure (as a whole) differs between groups. Unlike univariate tests, it takes into account interactions between all taxa, not just individual taxa.
  • Microbiome data are often not normally distributed, and PERMANOVA does not require this.

Regarding LEfSe, the assumptions are as follows:

  • The analysis does not take into account correlations between samples.
  • LEfSe uses sampling methods that must be normalized.
  • The method works with unpaired and paired samples, but special settings are required for paired data.

LEfSe identifies taxa that differ statistically significantly and biologically relevant between groups. The method not only estimates the significance (p-value), but also the effect size (LDA score), which allows to identify the most important taxa. LEfSe analyzes not only taxa, but also their classification (e.g. phylogenetic levels: kingdom, class, genus).

To conclude, these two methods complement each other well. PERMANOVA answers the question: "Is the microbiome the same between groups?" LEfSe answers the question: "Which specific taxa are responsible for the differences?"

To address the reviewer’s comment, the following statements have been added to the Methods section:

Raw read counts were converted to relative abundance (proportions) for community composition analysis. For alpha/beta diversity metrics, we performed rarefaction to the minimum sequencing depth (1,000 reads/sample) to ensure equal sampling effort across comparisons.

“PERMANOVA (Adonis) was used to assess differences between groups, taking into account interactions between all taxa.”

“Linear Discriminant Analysis Effect Size (LEfSe) identified bacterial taxa that differ significantly and biologically relevant between groups(Segata et al., 2011), esti-mating both the significance (p-value), but also the effect size (LDA score), which al-lows to identify the most important taxa”

Comment: Presentation of Figures: Ensure all figures, especially complex plots like LEfSe cladograms, are legible when printed. Improve label clarity in the cartoon summary (Figure 6C). Refer to each figure in the main text at the first relevant mention to help guide the reader.

Response: We appreciate the feedback regarding the clarity and readability of our figures. This important point was also raised by the Reviewer 2. We have carefully reconsidered all graphical presentations and implemented the following comprehensive improvements to enhance accessibility of our results:

  1. All axis labels, legends, and annotations have been enlarged across all figures, in addition, we increased font sizes for statistical markers (p-values) and taxonomic labels;
  2. We removed excessive white space through tighter figure margins, and rebalanced panel sizes to prioritize data visualization over empty space;
  3. We divided the original Figure 6 into two separate figures (now Figure 6 and Figure 7). This allows each figure to occupy full-width display with larger graphical elements.

We believe these changes substantially improve the reader experience and we're grateful for the opportunity to enhance our visual presentation. Please find the updated figures in the revised submission.

Comment: Interpretation and Discussion: Expand the discussion on how the Russian cohort context may affect microbiome profiles and the generalizability of findings. Add more depth when interpreting why cancer seems to mask the microbiome effects of smoking. Clarify the biological relevance of correlations (e.g., Selenomonas and smoking intensity) with hypotheses for mechanism if available.

Response: Thank you for this valuable suggestion. To address the reviewer’s point, we added the following statements to the Discussion section:

Our study is the first comprehensive characterization of respiratory microbiota in a Russian population, comparing healthy individuals and lung cancer patients across smoking status groups. The Russian context may influence microbiome profiles through several unique factors: the prevalence of particular tobacco products (including higher consumption of distinct local blends), environmental exposures (such as extreme seasonal climate variations and urban air pollution patterns in major cities), and population-specific lifestyle factors including dietary habits and antibiotic usage patterns.”

“In patients with lung cancer, microbiota diversity between smokers, former smokers, and nonsmokers was minimal. This suggests that tumor-associated factors may dominate over smoking history in shaping microbial communities, likely because the systemic inflammatory and immunomodulatory effects of malignancy exert a more profound influence on the respiratory microbiome than smoking-related changes alone. This phenomenon aligns with emerging evidence that cancer creates a pervasive microenvironment that fundamentally alters host-microbe interactions across anatomical sites.”

Comment: Sampling and Methodological Details: Provide more detail on the demographic matching between control and patient groups. Discuss potential contamination of sputum samples by upper airway microbiota, especially since this is a known limitation.

Response: Thank you for the comment. Please note that potential contamination of sputum samples by upper airway microbiota was discussed and mentioned as a limitation in the original version of the manuscript:

This study has several limitations. Whereas the sputum analyzed in our study serves as a surrogate for the LRT microbiota, it may still be contaminated with bacteria from the upper respiratory tract during sample collection(Pu et al., 2020)”.

While we acknowledge that matching demographic characteristics between cases (lung cancer patients) and controls (healthy subjects) can be methodologically ideal, we intentionally did not implement strict matching criteria for this study due to practical considerations in clinical research. Recruiting perfectly matched controls would have significantly prolonged our study timeline (estimated additional 12-18 months based on our center's patient demographics), delaying potentially impactful findings for lung cancer diagnostics. To address the reviewer’s concern, we added the following statement to the Methods section:

Demographic matching between cases (lung cancer patients) and controls (healthy volunteers) was not implemented due to practical recruitment constraints.”

Minor Comments:

Comment:  Abstract: Briefly mention key statistical methods and highlight the inclusion of former smokers as a strength.

Response: Thank you for your question. Please note that inclusion of former smokers was already in the abstract of the original manuscript:

“…the lower respiratory tract microbiome in a Russian cohort of 297 individuals, comprising healthy subjects and lung cancer patients of different smoking statuses (current smokers, former smokers, and nonsmokers)” 

To address the reviewer’s comment, we added the following statement to the Abstract: 

PERMANOVA (Adonis) test and Linear Discriminant Analysis Effect Size were used for statistical analysis of data.

Comment: Line 20: Change “Here, we characterize…” to “We characterized…” for consistency in tense.

Response: Done.

Comment: Line 259: Rephrase “may not fully restore microbiome diversity” to clarify timeframe/context.

Response: We changed the phrase may not fully restore microbiome diversity” to “may not be sufficient to reshape microbiome diversity” for clarity.

Comment: Review references for consistency in formatting (some DOI entries are inconsistent).

Response: Thank you for your comment. All the references have been reformatted to comply with the style of Cancers.

Comment: Consider mentioning the sequencing depth and whether rarefaction was performed.

Response: To address the reviewer’s point, we added the following statements in the Methods section:

Average sequencing depth was 11431.”

 “For alpha/beta diversity metrics, we performed rarefaction to the minimum sequencing depth (1,000 reads/sample) to ensure equal sampling effort across comparisons.”

We sincerely thank the reviewer for their constructive comments which greatly improved our manuscript!

Round 2

Reviewer 2 Report

Comments and Suggestions for Authors Thank you for the very thorough, thoughtful and detailed responses and in flawless English. Among the most comprehensive and well written responses I have ever received to my reviewer suggestions.

Reviewer 3 Report

Comments and Suggestions for Authors

No additional comments at this point.